# A Transmissive Theory of Brain Function: Implications for Health, Disease, and Consciousness

Nicolas Rouleau [1,2,*] and Nicholas Cimino [1]

1   Department of Psychology, Algoma University, Sault Ste. Marie, ON P6A 2G4, Canada
2   Department of Biomedical Engineering, Tufts University, Medford, MA 02155, USA
*   Correspondence: nicolas.rouleau@tufts.edu

**Abstract:** Identifying a complete, accurate model of brain function would allow neuroscientists and clinicians to make powerful neuropsychological predictions and diagnoses as well as develop more effective treatments to mitigate or reverse neuropathology. The productive model of brain function, which has been dominant in the field for centuries, cannot easily accommodate some higher-order neural processes associated with consciousness and other neuropsychological phenomena. However, in recent years, it has become increasingly evident that the brain is highly receptive to and readily emits electromagnetic (EM) fields and light. Indeed, brain tissues can generate endogenous, complex EM fields and ultraweak photon emissions (UPEs) within the visible and near-visible EM spectra. EM-based neural mechanisms, such as ephaptic coupling and non-visual optical brain signaling, expand canonical neural signaling modalities and are beginning to disrupt conventional models of brain function. Here, we present an evidence-based argument for the existence of brain processes that are caused by the transmission of extracerebral, EM signals and recommend experimental strategies with which to test the hypothesis. We argue for a synthesis of productive and transmissive models of brain function and discuss implications for the study of consciousness, brain health, and disease.

**Keywords:** brain function; transmission; electromagnetism; ultraweak photon emissions; consciousness

"If you wish to upset the law that all crows are black, you mustn't seek to show that no crows are; it is enough if you prove one single crow to be white."—William James

## 1. Introduction

*What do brains do?* For thousands of years, the brain has been regarded as the source of an individual's thoughts and actions and many cultures have claimed the brain as the seat of the soul or the location of a person's true *essence* [1–3]. Indeed, the experience of an observer existing behind one's eyes has likely been shared among members of the human species for at least hundreds of thousands of years. Some of our most ancient medical records indicate a longstanding knowledge of the impact of brain injuries on self-awareness, voluntary movement, and speech [4]. Our distant ancestors understood the significance of the brain and its value to the individual as evidenced by the development of helmets and rudimentary neurosurgical techniques such as trepanation [5,6]. While considerable technological advances over the past century have clarified the underlying mechanistic details, our general model of brain function remains largely unchanged. That is, modern neurologists view the brain as a thought-generating, memory-storing, and body-controlling organ. While none among them would deny the influence of the environment on the brain's function, its preeminent status as an object from which all psychological phenomena originate is generally accepted without question.

### 1.1. The Productive Model of Brain Function

Central to a contemporary understanding of brain function is the assumption that all features of cognition and behaviour—including all thought, memory, personality, and experience—can be reduced to and originate with the activities of cells, circuits, networks,

and other neural aggregates. In the field of neuroscience, few assumptions are less controversial than this *productive model* of brain function, which identifies the causal locus of thought as fundamentally internal. Indeed, the productive model of brain function benefits from its convergence with other bottom-up approaches in biology where the coordination of simple modules explains complex function. Physiological events at the cellular level are generally assumed to cause tissue-level dynamics, which in turn cause organ function. Applied to the brain, the productive model predicts that specific neural activations at the micro-scale should be highly correlated with psychological phenomena at the macro-scale and that stimulation or ablation of neural substrates should induce or suppress function, respectively. It also predicts that all brain function will be irreversibly lost upon death as the modules that produce cognition and behaviour gradually decay.

The model's success is reflected in over a century of empirical support. As predicted, patterns of neuronal activity reliably precede or co-occur with cognitive and behavioural functions [7]. When the same brain regions are stimulated or lesioned across individuals, outcomes are highly conserved [8–11]. Systematic manipulations of brain chemistry and electrical activity under laboratory and clinical conditions reliably alter cognitive states and response patterns. Viewed through the lens of a *productive model* of brain function, these data suggest that biological events within the brain are the proximate causal determinants of psychological phenomena. The corollary is that all brain functions can be traced back to the release of neurotransmitters across the synapse, the generation of action potentials, the activation of ion channels, and other micro-scale, biological events. However, this is only one of many possible interpretations of the data, which are often based upon correlation. Despite its many benefits and successes, the productive model suffers from some limitations. As we will discuss, several types of commonly reported psychological phenomena are reasonably dismissed due to their mechanistic implausibility under a productive model. A rigidly productive model also struggles to adequately address longstanding observations that brains respond to and cohere with electromagnetic (EM) signals in the environment, placing an undue burden on neural tissue as the lone generator of all mental events. Therefore, it may be worth considering the possibility that the productive model is not a complete model of brain function. To further explore its limitations and alternatives, we will ask the following question: *Are all brain functions fundamentally productive?*

Even the most advanced functional neuroimaging devices cannot establish causal certainty about the relationships between discrete neural events and co-occurring behavioural responses [12]. It is also worth noting that most neuroimaging techniques do not measure neural activity directly. Instead, activations are inferred by measurement of blood (de-)oxygenation, glucose metabolism, and other correlates [13]. Similarly, the suppression or expression of behaviour following an experimental brain lesion cannot necessarily be attributed to the damaged region itself [14]. Some consideration must always be given to the possibility that nodes upstream or downstream of the lesion have been inhibited or activated by disruption of the wider network and that these recondite nodes explain the observed variance. It is only by adopting a productive model of brain function that investigators feel sufficiently emboldened to derive causal relationships from ambiguous data. As empirically-derived mechanisms underlying specific brain functions continue to accumulate in the literature, it is becoming increasingly tempting to disregard the possibility that recondite factors may be explaining what appear to be straightforward cause-and-effect relationships between neural activations and their cognitive or behavioural correlates. As a general rule, this is a reasonable approach to scientific investigation because all data must be interpreted to derive meaningful or useful conclusions and established mechanisms are considered more plausible sources of explanation than hypothesized or novel mechanisms. However, in pursuit of knowledge, it is also reasonable to examine alternative interpretations of the data—especially when the established model struggles or fails to address key observations.

*1.2. The Transmissive Model of Brain Function*

In the late 19th century, William James described the productive model as a consensus position among psychologists and physiologists alike, despite the existence of alternative models of brain function with similar explanatory power [15]. He proposed a *transmissive model* of brain function, which positioned the brain as a system that sieves or filters rather than produces cognition. James drew on the analogy of the prism to explain how the brain might filter consciousness: a glass prism does not create a spectrum of colour from white light—it passively filters light, splitting the signal into an array of waves that exist independent of the prism [15]. The transmissive model was appealing to James as a means of extending consciousness beyond bodily death, since the living brain, like a radio receiver, captures and amplifies rather than creates its functional output state [16]. Unlike a productive brain, a transmissive brain exists independent of the signals it receives and therefore is not necessary for the existence of the signals. It would follow that damage to a transmissive brain would alter consciousness, but as a function of a misexpression of an intact signal rather than a failure to generate patterned activity from distorted tissues. A transmissive model of brain function would also provide a scientific framework with which to explain phenomena that were—and still are—reasonably marginalized by the universal adoption of a productive model of brain function. These would include widely-reported psychological and social phenomena [15,17], including premonitions [18], the experience of sharing thoughts with others [19], and historical cases of zeitgeists [20]. Interestingly, there is no reason to suggest a system cannot be both productive and transmissive in part; therefore, it may be possible to amend our current model of brain function instead of replacing it with a radical alternative.

To determine whether an amendment is necessary, we must ask ourselves: *Is there any evidence to suggest brain function is transmissive?* As we will demonstrate, there is considerable evidence indicating that brains receive, filter, process, and emit EM and optical signals [16]. Therefore, dozens of candidate mechanisms exist with the potential to order brain activity by transmission of information to and from the brain. If any subset of brain function is explained by causal elements outside the head that can impinge directly upon neural tissues, brain function cannot be fully explained by a productive model. In other words, and to paraphrase William James once more, if you wish to upset the law that all brain functions are productive, it is enough if you prove one single function to be non-productive. The goal of this paper is to make an evidence-based case for the existence of a limited subset of brain functions that are generated by transmission of extracerebral signals.

## 2. Properties of a Transmissive Brain

A transmissive brain will have receptive properties that allow it to interact with stimuli whose sources exist outside the body in the surrounding environment [15,16]. The initial stimulus must not pass through a sensory pathway—instead, it must be directly transduced by a physical mechanism involving brain tissue. The receptive structures may be conventional, such as molecular complexes (e.g., membrane-bound receptors), but may also be material-like (e.g., metals, gases, interfacial water), antenna-like (e.g., resonance of cell bodies, layered structures), or entirely novel. The net effect of transmission may be a generation or modulation of neural activity that can be localized to a particular brain region or generalized across multiple regions. Following the initial transduction, second order signaling may become integrated with classically productive functional mechanisms.

The canonical brain signaling pathways associated with the productive model include synaptic and electrotonic signaling [21,22]. In both cases, the electrochemical stimuli are local and brain-derived. For example, neurotransmitters are synthesized, transported, released, and ultimately sequestered by neurons as a form of cellular communication. The information within the system—biomolecules and patterned action potentials—can be said to have originated within the brain itself. While its true that brains acquire sensory data from their environments and receive molecular precursors to neurotransmitters from nutrients sourced from outside the body, the resulting signals—the proximate causes—are

internally generated. However, if a signal was able to pass through the skull and meninges to interact directly and systematically with brain tissue, it would meet the candidacy threshold for transmissive function. Interestingly, John C. Eccles—whose Nobel Prize-winning work helped characterize the neuronal action potential—proposed the existence of a field composed of particles he called "psychons" that could potentially transmit extracerebral information to neuronal dendrites to affect memory and consciousness [23,24]. Although the psychon has never been measured, its defining properties align with the transmissive model of brain function.

While it is possible that measurement of transmission will involve the identification of a novel particle or force, existing forces are likely sufficient. EM and optical signals are well-suited as messengers for transmissive communication. Indeed, both can pass through the skull and interact with brain tissues via known mechanisms while maintaining information fidelity across many scales of time and space [25,26]. Consequently, there are several existing theories of brain function that share a common EM basis and have historically been developed to address challenges in consciousness research. For example, E. Roy John's field theory of consciousness suggests that coherent resonance between the brain's EM field and quantum processes could synchronize neuronal firing—particularly, cortico-thalamic reverberations, which are frequently cited in the literature as integral to the generation of consciousness [27,28]. Similarly, Johnjoe McFadden's conscious electromagnetic information (CEMI) theory describes endogenous EM field modulations within the brain as a means of solving the binding problem and, in particular, the combination of independent percepts into a single experience [29–31]. Even the orchestrated objective reduction (Orch OR) model, developed by Hameroff and Penrose, is arguably an EM field theory, given its dependence on sub-cellular electronic interactions with microtubules, which display electric dipoles and readily align with electric fields [32–34]. Notably, Persinger developed a quantitative model that may account for several memory- and consciousness-based phenomena as products of EM field interactions with the brain [35,36]. Together, these theories point to EM fields as a source for brain–environment interactions that can account for transmissive function. However, mechanisms underlying said interactions must be empirically demonstrable if the transmissive theory of brain function is to survive scrutiny.

## 3. The Brain Is an Electromagnetic Organ

EM fields and light can directly modulate neural activity and are constantly emitted by brain tissues [16,37]. This should be unsurprising to most students of neurophysiology because, at a fundamental level, the brain is an electromagnetic organ—a fact admitted by even the most extreme version of the productive model of brain function. Supporting the role of EM as the basis for transmissive brain function (Figure 1), we will provide a brief review of the literature on the brain's sensitivity to and ability to emit EM signals.

### 3.1. Endogenous EM Fields of the Brain

The brain is partly composed of nearly one hundred billion electrically polar neurons, each discharging EM pulses in the form of action potentials dozens of times per second with detectable electric and magnetic field signatures [38]. For nearly one hundred years, electric fields associated with synchronized neuronal activity have been measured as voltage fluctuations at the surface of the scalp using electroencephalography (EEG) [39]. On average, the brain generates electric fields with intensities of approximately 2 mV/mm [40], which is sufficient to shift neuronal membrane potential by 0.1 to 1.3 mV [40,41] and modulate brain activity when applied at low frequencies (<2 Hz) [42]. Interestingly, neurons are also responsive to submillivolt per millimeter electric fields within the range of 300 μV/mm [43], which is significantly below the ambient electric field of the brain. It is therefore unsurprising that neuromodulation techniques, such as transcranial electric stimulation (TES), generate similarly intense (0.5 to 1 mV/mm) electric fields within superficial neural tissues at 1 mA stimulation currents [44]. While the brain's electric fields may be weak compared to those generated between the surface of the Earth and the ionosphere (~100 mV/mm), electric

fields of the brain are complex, highly ordered, and can be classified into spatiotemporal categories of "microstates" and "electomes", which are fingerprint-like EM patterns that are predictive of cognitive states, senescence, and neuropathology [45,46].

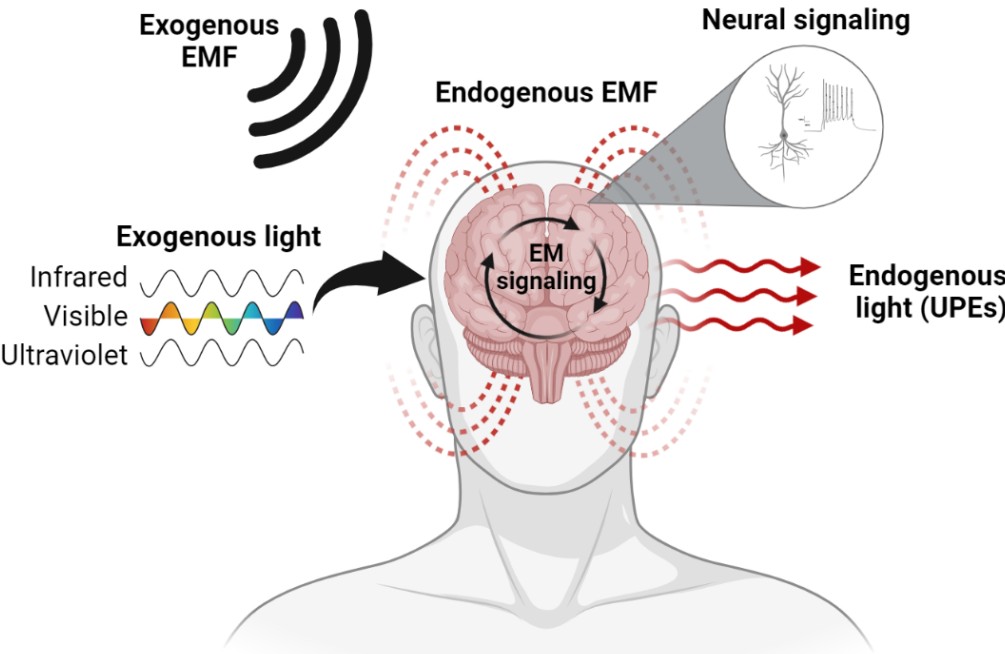

**Figure 1.** A schematic representation of the proposed EM-based transmissive model of human brain function. Endogenous and exogenous sources of EM fields (EMF) and light (including UPEs) are represented. EM signaling and canonical neural signaling co-occur and interact within the brain, synthesizing productive and transmissive mechanisms. Created with BioRender.com.

More recently, investigators have started to measure the brain's magnetic fields using a technique called magnetoencephalography (MEG) [47]. Magnetic field strengths of approximately 5 nT have been detected immediately adjacent to pyramidal cells [48]. At a distance of 1 mm from the surface of the neocortex, the brain displays bulk magnetic field intensities between 25 and 100 nT, which rapidly decays as a function of distance, detected within the pico- to femtotesla range outside the skull by MEG [25,49]. Therefore, the normal range of endogenous magnetic fields within brain-space is approximately 5 to 100 nT, which is orders of magnitude less intense than the Earth's magnetic field (~50,000 nT). Interestingly, MEG has been used to detect what are now the earliest signs of synchronous activity within the brains of developing fetuses [50,51]—an exciting discovery that may soon enable investigators to detect the elusive transition state from non-conscious matter to conscious brain tissue. Therefore, it may soon be possible to experimentally test the validity of panpsychism, the view that all matter is at least minimally conscious [52].

One of the most disruptive discoveries in neuroscience that has yet to be fully appreciated was the identification of a third neural signaling modality: ephaptic coupling [53]. Originally demonstrated by Katz [54] as increased excitability in cells located in parallel and adjacent to stimulated nerve fibers, the phenomenon was largely overlooked in the field until recently [55]. Unlike the hyperlocal connections between cells that define synaptic and electronic communication, ephaptic coupling is analogous to a wireless connection, where cells respond to and interact with their neighboring cells' EM fields [56]. It has been implicated as an important process for excitatory and inhibitory signaling within key areas of the brain including the olfactory regions, the cerebellum, and the hippocampus [56–58]. Slow, periodic, self-propagating waves of activity within the hippocampus were found to have been driven by ephaptic coupling and may be intimately involved in memory processes [58]. Because a cell's oscillating electric fields can likely constructively or

destructively interfere with those of their neighbors, there is a significant potential for non-directional, non-linear, and non-local activations. Electric fields associated with ephaptic coupling with intensities as weak as 0.5 mV/mm are capable of modulating and entraining neural oscillations [51,59]. Because ambient electric fields within brain-space display average intensities of 2 mV/mm, there is significant potential for endogenous feedback loops. These abundant ephaptic couplings, whose wireless connections may outnumber their wired counterparts, likely provide the brain with an enormous supply of virtual, looped connections that may facilitate reentry [60]—a process of reciprocal information flow that is considered important for consciousness.

The mechanism underlying ephaptic coupling is thought to involve EM induction, where the wire-like geometries of axons and other neurites generate electromotive forces in the presence of a magnetic field [53,61]. Anastassiou et al. [53] suggests that magnetic fields with intensities as weak as 5 nT may contribute to ephaptic coupling and some authors have implicated the magnetic fields of glial cells as modulators [61]. However, alternative EM-sensing mechanisms among biological organisms have been reported in the literature. Indeed, magnetoreception has been identified in bacteria [62], honeybees [63], pigeons [64], fish [65], bats [66], and humans [67] among other animals. Magnetoreception is thought to underlie migratory behaviour in several species [68]. Another proposed mechanism that was originally described and developed by Kirschvink [69] involves the detection of EM fields with biogenic magnetite, which are precipitated particles of iron oxide that can become permanently magnetized when exposed to an EM field. Although the precise interaction between magnetite and brain tissue is not fully understood, deposits of magnetite have been discovered throughout the human brain with localized, high concentrations found in the hippocampal bodies [70,71]. The hippocampus plays an integral role in memory formation and contains cells that encode and respond uniquely to different environments [72,73]. Its major input, the entorhinal cortex (EC), is a critical region associated with navigation in space and contains grid-like patches of tissue that guide the individual's movement through an environment [74]. The medial EC also contains high concentrations of pace-making stellate cells [75] that maintain hippocampal theta (~7 Hz) rhythms, which are essential for the temporal organization of movement, spatial navigation, memory encoding, and aspects of learning [76]. Interestingly, magnetite's ferrimagnetic properties may allow it to display magnetic hysteresis [77], which is a material property that can encode and maintain EM states as a form of memory [78].

### 3.2. Endogenous Light Emissions of the Brain

Photons are energetic particles of light that carry the EM force. Of course, nervous tissues, such as the retina, can detect light by dint of specialized photoreceptors, imbuing humans and other organisms with a capacity for vision; however, it is a lesser-known fact that brains can both transduce and emit light directly and independently of auxiliary sensory systems [79]. The human brain emits weak pulses of light with wavelengths within the ultraviolet, visible, and infrared ranges of the EM spectrum [80,81]. Ultraweak photon emissions (UPEs) are distinct from bioluminescent light emissions and are linked to the intrinsic biochemistry of cell metabolism (e.g., reactive oxygen species) and microtubule dynamics [82]. The physiological role of brain biophotons is not yet clear; however, effects involving aging [83] and cognitive capacities [80] have been reported. Seminal experiments linking neurotransmission and UPEs identified that brain tissues stimulated with aliquots of glutamate markedly increased their light emissions [84]. It was recently proposed that brain-derived light emissions could serve as information carriers for optical neurotransmission [85]. As originally predicted several decades ago [86], evidence is now indicating that axons may conduct and guide light like fiber optic cables [85], potentially increasing the speed and information density of neural communication by many orders of magnitude. Because information can be encoded within the direction, amplitude, frequency, and rotational parameters of photons [87,88], the possible existence of a complex optical neural network superimposed upon the canonical signaling pathways is an exciting new frontier

for neuroscience. Since white light can travel through the skull [26], UPEs may also provide a mechanism for transmissive, brain-to-brain communication.

If the brain does use light as a signal for communication, it should display photoreceptive properties independent of the visual pathway. Remarkably, within the darkened environment of the skull, there are many brain regions that respond to light within the visible and near-visible EM spectra. The cerebral cortices, hypothalamic nuclei, the striatum, the pineal organ, and other regions express photoreceptors, including non-visual opsins, such as Opsin3 or encephalopsin [89,90]. Most isoforms of encephalopsin absorb blue-green light with wavelengths below 500 nm and, when stimulated, can inhibit neural activity [91]. Some non-visual opsins, such as Opsin5, can detect ultraviolet light [92]. The widespread distribution of photoreceptors in the brain suggests a functional role; however, it remains unclear how they affect human cognition and behaviour. Investigations of non-visual opsins in non-human animal brains suggest potential effects on mood, endocrine function, circadian rhythms, and phototaxis even after enucleation [79]. Because brain tissues can emit wavelengths within the receptive range of Opsin3 and other non-visual opsins [93], endogenous optical signaling within the brain may be possible. If light from environmental sources, including other brains, can stimulate these optical signaling pathways, transmissive functions can be experimentally demonstrated.

### 3.3. EM–Brain Interactions in the Laboratory

Consistent with their endogenous EM characteristics, brains readily respond to EM field and light exposures in controlled, laboratory settings. At the microscopic level, neurons have been observed to align with and migrate along electric fields (galvanotaxis) in vitro [94]. EM fields can also modulate neuronal activity by affecting calcium flux and microtubule dynamics [95,96]. In humans, applied EM fields can treat disorders of brain function [97–99], alter perception [100], and modify behaviour [100], as well as evoke forced movements that are paired with a false sense of free will [101]. Some of the more commonly reported experiences during exposures to the intense (>1 tesla) EM fields associated with magnetic resonance imaging (MRI) include perceived flashes of white light (phosphenes), a metallic taste, headaches, and nausea [102]. Most symptoms can be explained by activation of the temporal lobes and their deep structures [103,104]. As described previously, induction is the assumed mechanism underlying these effects. Similarly intense fields are involved with applications of transcranial magnetic stimulation (TMS) to treat depression and anxiety among other psychiatric disorders. Rhythmic TMS applications are associated with long-lasting entrainment of brain oscillations [105]. Much weaker fields (microtesla range) with complex pulse patterns have been used to treat symptoms of brain injury [106], modulate seizures [107], and inhibit the experience of pain (analgesia) [108], as well as induce out-of-body and sensed presence experiences [109–112]—both of which are spatial in nature. Among the weakest bioeffective magnetic fields are within the nanotesla range, mirroring physiological intensities. Indeed, several studies have confirmed that applied magnetic fields with intensities ranging from 5 to 100 nT can impact neural activity and development [113–116].

In addition to EM fields affecting brain activity, there is considerable empirical evidence that neurons can be activated or deactivated by non-visual stimulation with artificial light sources. It is important to note that none of the following effects are associated with optogenetic actuators or the use of photosensitizing molecules; wild type brain tissues naturally display response patterns consistent with photoreception. Indeed, rhythmic light stimulation can entrain brain oscillations in real-time and transcranial infrared light can modulate EEG alpha power [117,118]. At the cellular level, artificial light exposures have been used to facilitate the release of neurotransmitters like glutamate and dopamine [119]. Mice exposed to transcranial light (400–500 nm range) via the ear canal displayed time-of-day- and brain-region-dependent effects on monoamine levels and Opsin3 expression, paralleling the regulation of receptors normally associated with repeated exposure to chemical compounds [120]. Similarly, a solid-state blue (475 nm) laser was used to inactivate neural

activity associated with auditory transmission in mice [121]. In the latter study, experiments with slice preparations revealed that light stimulation photobleached native flavoproteins, thus inhibiting the aerobic energy metabolism upstream of neuronal activity [121]. Interestingly, small-diameter axons are selectively, rapidly, and reversibly inhibited by infrared light compared to large-diameter fibers [122]. Together, these data indicate that light across the visible and near-visible EM spectra can be used to experimentally modulate brain activity. The question of how environmental stimuli could recapitulate these effects in the naturalistic setting and the implications for cognition remain largely unaddressed.

### 3.4. EM–Brain Interactions in Naturalistic Settings

Even if some brain function can be explained by transmission rather than production, the relevance of a transmissive model is contingent upon the existence of signals within everyday environments that can systematically interact with brain tissues. The most pervasive EM field on Earth is generated by the rotation of molten iron within its core [123]. At the surface of the planet, Earth's magnetic field intensity is approximately 50 µT (or 50,000 nT) on average [124]; however, it varies over time [125] and by latitude [126]. The intensity of Earth's electric field is approximately 100 V/m (or 100 mV/mm) [127]. Both are frequently disturbed by solar wind and coronal mass ejections [128], which can transfer increased EM energy to the magnetosphere. *Are Earth's EM fields sufficiently intense to affect brain activity?* As discussed previously, both the electric and magnetic components of the geomagnetic field are sufficiently intense to activate neural tissues. Further, over a century of research indicates that natural EM–brain interactions are common and may exacerbate select neuropathologies. Indeed, reports from the early 20th century reveal that psychiatric hospital admissions were significantly correlated with fluctuations in geomagnetic activity [129]—a finding that has seen been independently confirmed [130]. Epileptic patients, whose brains are characteristically excitable, are particularly sensitive to geomagnetic activity, which correlates with the frequency of their convulsions [131]. It is worth noting that most seizures originate with the temporal lobes [130], which may be particularly sensitive to EM fields, as discussed elsewhere. Recent evidence indicates that human magnetoreception may be dependent upon the wavelength of ambient light [132], suggesting a potential role of non-visual optical brain communication. These convergent observations suggest EM–brain interactions are not only possible but likely clinically significant.

Interestingly, brain activity and the Earth's EM field share several properties, including operating frequencies. Due to the consistent frequency of global lightning strikes that occur between the ionosphere and the Earth's surface every second (~40 Hz) [133], the geomagnetic field displays an intrinsic oscillation with a frequency mode of approximately 7.83 Hz [134]. This resonant frequency or "Schumann resonance", originally described by W.O. Schumann, is also associated with harmonic frequencies of approximately 14, 20, 26, and 33 Hz [134]. Incidentally, it is the unique pairing of 7 Hz "theta" activity and 40 Hz "gamma" activity within the hippocampal bodies that allow the brain to encode and store memories associated with conscious experience [135]. Since the discovery of Schumann resonance, several independent investigators have observed marked similarities between electrophysiological recordings of brain activity and simultaneous fluctuations of geomagnetic activity [136]. Building on the seminal findings of König and colleagues, recent investigators have confirmed that EEG oscillations can become coupled with geomagnetic fluctuations. Specifically, brain oscillations within the theta (4–7 Hz) and alpha (8–13 Hz) EEG bands can become synchronized with the Earth's EM field [137]. Saroka and colleagues [138] were the first to demonstrate real-time coherence between Schumann resonance and the frequency spectra of EEG rhythms across hundreds of independent human brains. Confirming the EM–brain interaction, Kirschvink's team recently described a series of experiments that demonstrate a significant, orientation-dependent desynchronization of alpha-band rhythms (8–13 Hz) associated with the static magnetic field of the Earth [137]. Interestingly, shielding the brain from EM fields with copper material can also selectively attenuate alpha rhythms [139]. It may be relevant that quantitative models of the

brain's intrinsic resonance frequency and material properties predict optimal receptivity to frequencies between 7 Hz and 10 Hz [140,141].

While all states of consciousness are likely affected by Earth's EM fields and hormones, such as melatonin, are tied to its variations [142], the sleep state is particularly vulnerable to changing geomagnetic activity. Frequent correlates include reports of vivid, bizarre, and prophetic dreams [143,144]. Orientation of the head during sleep relative to magnetic North is a determinant of the duration of rapid eye movement (REM) sleep [145,146]. There may even be a relationship between geomagnetic activity and sleep paralysis [147]. Interestingly, sleep spindles—bursts of synchronous neural activity measured by EEG during non-REM sleep [148]—display frequency spectra (11–15 Hz) that overlap with geomagnetic harmonics. Together, these data indicate EM–brain interactions are likely possible and probably more integrated with our normal physiological states than currently assumed. Other psychological correlates of geomagnetic activity include mood, aggression, and several perceptual phenomena [149]. A transmissive model of brain function based upon EM signaling would position the brain as a receptive organ whose cognitive and behavioural correlates are at least partly determined by EM perturbations in the environment.

## 4. Brain Tissue as an EM-Receptive Biomaterial: A Hypothesis

EM–brain interactions are most associated with synchronous brain rhythms within the theta–alpha band with a temporal lobe and/or right hemispheric focus [139,150,151]. Whether the brain is exposed to natural EM field perturbances, experimental simulations, or is shielded from exogenous EM, it generally displays this peculiar selectivity. *Does the productive model of brain function offer any insights when interpreting these data?* Commonly cited mechanisms underlying neural magnetoreception include magnetite-based sensing and the activation of select ion channels; however, independent of the precise mechanism, there is little doubt that EM fields can interact with neurophysiological processes. A more conventional explanation may implicate the medial EC's stellate cell population as a likely candidate for EM-receptive function, given its role as a 7 Hz (theta) pacemaker. Additionally, magnetite is highly concentrated within the neighboring hippocampus relative to most other brain regions. However, experiments with ex vivo brain tissues suggest theta rhythms are preferentially amplified by the same temporal regions post-mortem [139,150,151]. Indeed, when post-mortem human brain specimens fixed in ethanol–formalin–acetic acid (72% ethanol, $C_2H_6O$; 5% formaldehyde, $CH_2O$; 5% acetic acid, $C_2H_4O_2$; 18% distilled water, $dH_2O$; pH = 2.9) were directly injected with current or exposed to time-varying EM fields, mesial temporal lobe structures (e.g., hippocampus, parahippocampal cortex, entorhinal cortex, uncus) were found to selectively filter the induced signals with notable 4 Hz–7.5 Hz (theta) and 7.5 Hz–14 Hz (alpha) amplifications [150,151]. Interestingly, when the same post-mortem brain specimens were exposed to naturally occurring geomagnetic storm conditions (EM field perturbations on Earth caused by solar activity), right parahippocampal alpha activity became markedly amplified relative to non-storm conditions [151]. These experiments suggest that EM–brain interactions may operate outside of canonical signaling modalities and may even be independent of living cell function since effects persist after chemical fixation and cell death.

One possible solution would involve the filtration of EM signals by a material interaction with brain tissues. Unlike canonical signaling mechanisms involving living cells actively processing electrical or chemical information, we hypothesize that brain tissue may passively interact with EM signals due to its intrinsic material properties. Quantitative modeling indicates that the permeability and permittivity of brain tissue predict a resonance frequency of approximately 7 Hz while bulk signaling within the spatial dimensions of the cerebrum predict 10 Hz [140,141]. Indeed, the finger-print-like, structural "identity" of each cytoarchitectonically distinct brain region may display a unique capacity to filter EM signals [139,150,151]. While largely speculative at this time, the brain's highly ordered, repetitive microstructure may provide the conditions for metamaterial-like interactions between EM fields and arrays of neural aggregates—particularly within the temporal

lobes. For example, the swiss-roll-like interlocking c-structures of the hippocampal–dentate gyrus interface or the neocortical–archicortical transition from 6- to -3 laminae may display intrinsic, EM-receptive material properties.

Perhaps material-based EM–brain transmissions increase the excitability of select neuronal populations, thus increasing the probability that they will be activated by productive signaling modalities. Which is to say, transmissive brain function may serve as a modulator of established neurophysiological processes. If the probability of a particular brain region's activation is determined by environmental factors, such as geomagnetic field perturbations, there is a strong argument to be made for a partly transmissive model of brain activity. Indeed, the major implication would be that thought and behaviour are subject to subtle extracerebral EM signals and their fluctuations. Just as light conditions alter mood and other features of cognition by productive mechanisms involving the visual pathways and their circadian or limbic offshoots, environmental EM fields and non-visual optical signaling may provide similar regulation. Of course, more experimental evidence will be needed to determine the validity of the proposed hypothesis.

## 5. Novel In Vitro Approaches to Test EM-Based Transmissive Function

To our knowledge, very few authors have explicitly discussed the possibility of transmissive brain function within recent decades [152–154] and the model has never been formally tested. Here, we have cited several lines of experimental evidence that indirectly support an EM-based mechanism for transmissive brain function. The data indicate (i) that brains emit EM fields and light, (ii) that similarly intense EM and optical stimuli can activate neural tissues, (iii) that these stimuli can originate from environmental sources, and (iv) that some of these activations are explained by material-like properties that are not contingent upon productive mechanisms of cells.

The most compelling empirical findings in support of EM–brain interactions that can potentially drive transmissive brain function involve in vivo experiments with humans that demonstrate synchronous neural activity inferred by EEG can be systematically manipulated by applied EM fields or by attenuating environmental EM fields using shielding devices, such as Faraday cages. However, there are some limitations to these experiments that warrant alternative approaches. Most importantly, the brains of living humans are continuously displaying productive functions, such as the generation of action potentials, which may obscure the impact of transmission. Without the ability to systematically activate and deactivate productive function, it may be impossible to uncouple intracerebral and extracerebral factors as contributors to brain function. Similarly, the material properties of brain tissues cannot be assessed in vivo since there is no control over cytoarchitecture and other physical parameters. The brains of animals, including humans, are not structurally tunable and only partially accessible by minimally invasive probes.

In vitro approaches may provide new insight in pursuit of EM-based mechanisms for transmissive brain function. Despite their own limitations—including reduced complexity, an absence of observable behaviour (i.e., they are disembodied), and an inability to assess subjective experience—in vitro models of the central nervous system address all of the aforementioned limitations. Specifically, they provide structural tunability, ease of access, and high tractability for maximal degrees of control. Indeed, bioengineered 3D brain models, including organoids, spheroids, and bioprinted and scaffold-based constructs, as well as brain-on-a-chip specimens, are quickly becoming the gold standard in neuroscience research as tools to recapitulate fundamental neural circuitry [155] as well as diseased states [156]. Using neural tissue engineering techniques, it may be possible to ask the question: *Can information be transmitted to neural tissues using EM and optical signals?*

To identify the role of cytoarchitecture on the capacity for neural tissues to filter EM signals or selectively amplify specific frequencies, 3D bioengineered tissues may be designed and assembled with several customizable parameters [156]. To model the neocortex in vitro, many factors can be customized, including cell types (e.g., densities of pyramidal cells, astrocytes, microglia), number of cortical layers, extracellular matrix composition

(e.g., concentrations of collagen IV, proteoglycans), and electrical conductivity (e.g., incorporating synthetic materials like PEDOT, PSS, carbon nanowires) [157,158]. Each tissue iteration could then be exposed to sweeps of EM field frequencies applied at physiological intensities while electrophysiological measurements are recorded as local field potentials (LFPs). Patterned EM fields designed to mimic Schumann resonance could also be used to test specific hypotheses regarding environmental sources for EM–brain interactions. Magnetite and other materials could be embedded within custom tissues to test specific hypotheses regarding the mechanism of magnetoreception. To assess the contribution of productive and transmissive mechanisms, tissues could be manipulated to transiently or permanently suppress endogenous activations. For example, measurement of EM-based signal filtration could be conducted with or without presence of tetrodotoxin and other chemical inactivators. Alternatively, the EM-filtering properties could be assessed before and after chemical fixation to assess active and passive contributors. The proposed methods would help identify cytoarchitectural determinants of EM–brain interactions; however, similar strategies could be used to assess light-based signaling.

## 6. Implications for Health and Disease

*How would the adoption of a transmissive model of brain function, in part or whole, impact the way we view brain health and disease?* Currently, neurological disorders are viewed as dysfunctions within biological substrates that produce disruptions of cognition and/or behaviour. Whether the dysfunction is explained by a genetic mutation, epigenetic modification, developmental defect, acquired brain injury, or by infection, the symptoms and signs are attributable to an internal dysfunction of the brain's "hardware" or "software". Thus, diagnosis is focused entirely on the brain as the generator of the dysfunction. Consistent with this productive model, contemporary treatments aim to regulate faulty signaling, repair damaged tissue, or otherwise correct a defect along the canonical signaling pathways that generate thought, experience, and behaviour. The productive model of brain function demands that all focus be directed toward the organ itself and its internal mechanisms. Through the lens of a transmissive model of brain function, all the underlying neurobiology would be identical; however, the ultimate explanations and underlying causal mechanisms would be fundamentally different.

Disease in the context of an EM-based transmissive model can only be accurately understood as an abnormal interaction between the brain and its extracerebral signals. For example, in the case of brain injuries, which may damage, distort, or annihilate neural substrates, downstream cognitive and behavioural effects may be attributable to the disruption of EM–brain receptivity, encoding, or processing. As brains degenerate, their ability to receive transmissions may diminish or become "out-of-tune" with environmental EM regulators. Thus, injury-related dysfunctions could be viewed as misexpressions of an intact extracerebral signal. Interventions would involve re-establishing transmissive capacities. Consider the analogy of the radio receiver: damaging the receiver may appear to alter the information content of the signal but, as any independent measurement of the EM signal would reveal, this is only illusory. This illusion is further reinforced when the information content is re-established upon repairing the receiver. A productive model of brain function, though technically inaccurate, would perfectly predict this pattern of effects. However, adopting the accurate, transmissive model would invite novel approaches to diagnosis and treatment. Symptoms such as forced thought, delusions of persecution, unidentified voices, and other psychiatric phenomena may require some reinterpretation. As an example of how approaches to treatment would change, it may be possible to introduce an intermediate receiver or processor, possibly in the form of a neural–computer interface, that compensates for misexpressions of transmitted signals due to brain-based disruptions of receptivity. Practically speaking, this may involve the use of brain implants to compensate for errors of transmission instead of attempting to fully regenerate lost brain tissues with the hopes of reinstating a productive function of the tissue itself.

In a similar way, neural ontogeny can be viewed as a gradual acquisition of transmissive capability—as brain tissues mature, cells differentiate, migrate, and build structurally stable cytoarchitectures, they become increasingly in-tune with EM-based signals in the environment as a normal feature of development. This interpretation would be consistent with the overrepresentation of EM-sensitive REM cycles and related rhythmic phenomena in infant sleep. While many questions arise from a transmissive revision of brain-based phenomena, the potential impact on mental health and medicine would be significant. As greater attention is being paid to the effects of natural environments on mental health [159], the transmissive model of brain function is more likely to be seriously considered.

## 7. Discussion and Conclusions

To restate the Jamesian principle: *If you wish to upset the law that all brain functions are productive, it is enough if you prove one single function to be non-productive.* Here, we have provided an evidence-based, scientific case for the existence of transmission-based brain functions as well as some guidance to experimentally assess its validity. While we do not reject the productive model, given the evidence provided here, it seems reasonable to amend our contemporary understanding of brain function to include the transmissive effects of extracerebral EM signals from environmental sources. We argue that the brain is both receptive to and actively emits EM signals. EM–brain interactions, verified and supported by experimental evidence and quantitative observations in naturalistic settings, provide plausible mechanisms for a transmissive model of brain function. The implications for the origins of consciousness as well as our interpretation of brain health and disease are significant. Investigators should design experiments to test the brain's transmissive properties directly while accounting for endogenous and exogenous sources of variance. Future studies may involve identifying EM-based signals that are selectively amplified or attenuated by brain tissues as well as parameters that affect filtration capacities. It may be necessary to design artificial brains in silico or in vitro with which to test the hypothesis that brain tissues are EM-receptive biomaterials. Three-dimensional bioengineered brain tissues provide optimal tractability as model systems with which to test EM-based transmissive function. Separating the contributions of active, living cells and passive, material-, or biological-antenna-based structures can be given special consideration with the use of tunable, in vitro neural tissues. If paired with well-controlled experiments in vivo, it may be possible to experimentally to isolate mechanisms of transmission. In pursuit of a more complete model of brain function, it may be necessary to synthesize productive and transmissive models.

**Author Contributions:** Conceptualization, N.R.; Project administration, N.R.; Visualization, N.R.; Writing—original draft, N.R. and N.C.; Writing—review & editing, N.R. and N.C. All authors have read and agreed to the published version of the manuscript.

**Funding:** This research received no external funding.

**Institutional Review Board Statement:** Not applicable.

**Informed Consent Statement:** Not applicable.

**Data Availability Statement:** Not applicable.

**Acknowledgments:** The authors would like to dedicate this paper to the late Michael A. Persinger, whose pioneering work inspired the development of the ideas outlined herein. They would also like to thank Nirosha J. Murugan for her indispensable insights and helpful suggestions, as well as for her critical and longstanding support. Additionally thanks are owed to K.S., M.B., L.T. and C.W. for their helpful insights and comments. Finally, we acknowledge the Bigelow Institute for Consciousness Studies (BICS) as a significant motivator and supporter of the work.

**Conflicts of Interest:** The authors report no conflict of interest, financial or otherwise.

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
