# Peer review of "A Transmissive Theory of Brain Function: Implications for Health, Disease, and Consciousness"

_neurosci, doi:10.3390/neurosci3030032_

Round 1
Reviewer 1 Report
The current manuscript attempts to challenge contemporary models of brain function/behaviour that argue neural processes as the sole causal factor for the generation of complex cognitive and behavioural states, such as consciousness. The alternative transmissive model proposed here expands on earlier theories, most notably proposed first by Henri Bergson and William James but later expanded by others such as Frederic Myers, Pierre Janet, Aldous Huxley etc., which argues that the brain does not directly produce consciousness, but instead, functions as a filter, channel, or amplifier of pre-existing consciousness phenomena that reside outside and all around us.
The authors provide a cogent well thought-out argument that is supported by strong empirical and theoretical data showing that the brain is highly sensitive and receptive to extracerebral electromagnetic signals as well as exogenous light sources that occur within the environment. The author further discusses how ephatic coupling and brain-derived photon emissions could function as plausible substrates to support the transmissive model of brain function. Importantly each of these candidate models could in theory be tested experimentally, which provides further support to the model being proposed here.
In general, this is an extremely interesting and thought-provoking paper. I do, however, have some very minor comments that I hope the authors could address.
1. Page 12 – Lines 83-85: The authors should expand on this point a bit more. It would be helpful if the authors discuss the benefits and strengths of the productive model. Clearly, the productive model must have some success as serves as the predominate theoretical framework for addressing brain-behavior relationships. I think it would be first helpful if the authors discuss what the benefits and successes of this model are and what the specific limitations or explanatory missteps that come from accepting this model. This will help presenting the alternative model discussed here, i.e. transmissive model.
2. Page 5 Line 218 “…ephaptic coupling would increase the number of connections by many orders of magnitude because every cell within the brain may have the potential every other cell”. This is a very interesting idea, but it also seems like a bit of a stretch to at the same time. Wouldn’t ephatic exchanges between neurons or adjacent nerve fibers be influenced by geometric considerations (e.g., distance and alignment of axonal processes) or by the high extracellular resistance thereby limiting the scope of their effects. I would further presume that the insulating effects of myelin might reduce probability of ephatic exchanges and prevent the type of meso/macro-scale interactions being proposed here. It would be important for the authors to provide some alternative explanations.
3. Page 7 Lines – 320-326. I understand the intention of the authors. However, the overall point that “God experiences” might be explained by environmentally drive EM-brain interactions is highly contentious and the overall idea is not well developed as it is unclear what specific environmental factors are being considered. In my opinion this overall point does not fit with the general theme and purpose of the paper and I recommend that the authors consider removing it.
Author Response
Please see the attached .docx with responses. Thank you.

Reviewer 2 Report
The manuscript focusses on the idea that the brain could be productive (all the brain functions originate in the brain) or transmissive (the brain acts as a radio receiver modulating and processing incoming information). The productive theory is held by most neuroscientists in the field and the transmissive theory is highly speculative at this time. The authors suggest that all is needed to show that the transmissive theory has some legs is to show that at least one brain function can be shown of the transmissive in nature. The authors discuss two potential mechanisms: ephaptic coupling and light induced neural activity. They also discuss magnetic fields as potential external influence on the brain.
The manuscript is well written and thorough. However, I find that there is little that is novel and enlightening as far as guiding new experiments. The ephaptic coupling paper cited (Anastassiou) as evidence for ephaptic coupling is fairly old and shows that although low amplitude electric field can influence the membrane voltage of neurons, the results show that that these applied field are not sufficient to activate neurons or neural activity. The fact that endogenous field can interact with the neural networks that generate them was published long ago (Endogenous Electric Fields May Guide Neocortical Network Activity
Flavio Fröhlich and David A. McCormick, 2010. The fact that magnetic field can interact with the brain is also known and no new mechanism is proposed. Large external magnetic can clearly affect neural activity through induced electric fields, but the reverse has not been shown to be true. Although neural activity can indeed generate detectable magnetic fields, those fields in turn have not shown to be capable of activating a neuron.
The authors do propose a hypothesis that the brain properties can play a significant role in the filtering of incoming external activity by acting as a filter of certain frequencies particularly in the low frequency range such as theta. This is not a new idea as many papers have focused on the resonance properties of the brain tissue and brain membrane. Moreover, if this property were to confer the transmissive properties to the brain, small but detectable electric fields applied near the head would clearly affect brain function. Although small amplitude tDCS or tACS currents applied to the skull can clearly affect neurons, electric fields at similar frequencies/amplitudes applied near the head do not produce any effects.
Therefore, I find that this manuscript lacks novelty in the ideas proposed and the authors do not propose new experiments that could shed light on this problem.
Author Response

(The authors gave the same response as above.)

Round 2
Reviewer 2 Report
The authors have made revisions and partially improved the manuscript and answered my comments. The fact that it is a review should be made clear. The transmissive hypothesis for the brain is not a new one. In fact it is quite old. Even after the revisions, the manuscript still does not add significantly to the knowledge in the field and I think it should be rejected.